# In-Season Body Composition Effects in Professional Women Soccer Players

**DOI:** 10.3390/ijerph182212023

**Published:** 2021-11-16

**Authors:** Rafael Oliveira, Ruben Francisco, Renato Fernandes, Alexandre Martins, Hadi Nobari, Filipe Manuel Clemente, João Paulo Brito

**Affiliations:** 1Sports Science School of Rio Maior–Polytechnic Institute of Santarém, 2040-413 Rio Maior, Portugal; rfernandes@esdrm.ipsantarem.pt (R.F.); alexandremartins@esdrm.ipsantarem.pt (A.M.); jbrito@esdrm.ipsantarem.pt (J.P.B.); 2Life Quality Research Centre, 2040-413 Rio Maior, Portugal; 3Research Center in Sport Sciences, Health Sciences and Human Development, 5001-801 Vila Real, Portugal; 4Exercise and Health Laboratory, CIPER, Faculdade Motricidade Humana, Universidade de Lisboa, 1499-002 Lisbon, Portugal; ruben92francisco@gmail.com; 5University of Trás-os-Montes e Alto Douro, 5001-801 Vila Real, Portugal; 6Department of Exercise Physiology, Faculty of Educational Sciences and Psychology, University of Mohaghegh Ardabili, Ardabil 56199-11367, Iran; hadi.nobari1@gmail.com; 7HEME Research Group, Faculty of Sport Sciences, University of Extremadura, 10003 Cáceres, Spain; 8Sports Scientist, Sepahan Football Club, Isfahan 81887-78473, Iran; 9Department of Physical Education and Sports, University of Granada, 18010 Granada, Spain; 10Escola Superior Desporto e Lazer, Instituto Politécnico de Viana do Castelo, Rua Escola Industrial e Comercial de Nun’Álvares, 4900-347 Viana do Castelo, Portugal; filipe.clemente5@gmail.com; 11Instituto de Telecomunicações, Delegação da Covilhã, 1049-001 Lisboa, Portugal

**Keywords:** phase angle, female, body fat mass, fat-free mass, intracellular water, rated perceived exertion

## Abstract

This study aimed to analyze anthropometric and body composition effects in professional soccer women players across the early and mid-competitive 2019/20 season. Seventeen players (age, height, body mass, and body mass index of 22.7 ± 6.3 years, 167.5 ± 5.6 cm, 60.7 ± 6.6 kg and 21.6 ± 0.2 kg/m^2^) from a Portuguese BPI League team participated in this study. The participants completed ≥80% of 57 training sessions and 13 matches. They were assessed at three points (before the start of the season (A1), after two months (A2), and after four months (A3)) using the following variables: body fat mass (BFM), soft lean mass (SLM), fat-free mass (FFM), intracellular water (ICW), extracellular water (ECW), total body water (TBW), and phase angle (PhA, 50 Khz), through InBody S10. Nutritional intake was determined through a questionnaire. Repeated measures ANCOVA and effect sizes (ES) were used with *p* < 0.05. The main results occurred between A1 and A2 for BFM (−21.7%, ES = 1.58), SLM (3.7%, ES = 1.24), FFM (4%, ES = 1.34), ICW (4.2%, ES = 1.41), TBW (3.7%, ES = 1.04). Furthermore, there were significant results between A1 and A3 for FFM (4.8%, ES = 1.51), ICW (5%, ES = 1.68), and PhA (10.4%, ES = 6.64). The results showed that the water parameters improved over time, which led to healthy hydration statuses. The training load structure provided sufficient stimulus for appropriate physical fitness development, without causing negative disturbances in the water compartments.

## 1. Introduction

Soccer is considered one of the most popular sports worldwide [1]. To improve soccer athletes’ performance and health, the assessment of anthropometric and body composition variables have been considered crucial [2]. Especially at a competitive level, body composition is an important component in an athlete’s fitness and health profile and in each sport, performance is improved in specific ways in order to prevent injury risk [3].

Special attention has been paid to body fat mass (BFM) and fat-free mass (FFM). It is well known that an increased fat mass compromises performance, while increased muscle mass can promote the development of strength and power, which are important for players’ performance [4,5,6]. According to a recent consensus statement, there are no values for BFM or FFM that should be followed, even more if we consider female soccer players [6]. For instance, in female player from US collegiate division 1, BFM of 16% was observed. In fact, the consensus statement added that it is not known what kind of body composition changes during the season may impact positively or negatively on the performance of the players [6].

Moreover, the interest in assessing other body composition variables, such as total body water (TBW), intracellular water (ICW), and extracellular water (ECW), to monitor hydration status in athletes has grown. For example, some studies have shown that ICW is a good predictor of strength and power in athletes [7,8,9].

Thus, considering the importance of body composition for athletes, frequent assessments should take place. This will allow coaches and athletes to know the development of body composition throughout the sports season and adjust training programs to prevent injuries and enhance sports performance.

Over the last decades, women’s participation in sports has greatly increased. Although scientific research on women soccer athletes is growing, it is still limited [5,10,11,12,13]. Coaches and sports-related professionals should be aware of gender-specific questions and needs for optimizing performance. Especially at an elite level, few data have been used to show changes in anthropometric and body composition in women soccer players during the in-season [14]. To the knowledge of the authors, if the variables mentioned above and the training load variables, such as rated perceived exertion (RPE), were considered simultaneously, no studies were found. According to a recent report, performance measured by training and/or match data and body composition assessment could help soccer coaches and their staff to provide proper information for each player [6].

Specifically, internal load, which is one of the two dimensions of load monitoring (the other is external load), is a crucial psychophysiological part of the training load monitoring processes. One of the most frequently used variables to access internal load is RPE or the session-RPE (s-RPE, multiplication of RPE by session duration). This measure is a valid, reliable, and sensitive approach to quantify and qualify the internal load while using a simple questionnaire [15].

Knowledge of the essential characteristics for successful women’s team soccer performance is useful to coaches, physicians, nutritionists, and exercise physiologists to improve their knowledge about women soccer athletes.

Therefore, this study aims to analyze the variations on anthropometric and body composition variables and their relationship with internal load in elite women soccer players across early and mid-competitive in-season using bioelectrical impedance analysis (BIA).

## 2. Materials and Methods

### 2.1. Experimental Approach to the Problem

This was an analytical and observational cohort study. The training sessions were performed during a five-month period, from September to January (early-to-mid-season) due to the COVID-19 pandemic, which provoked the disruption of training sessions and matches and the suspension of the season in March. The anthropometric and body composition assessments were conducted on three different occasions: the first week of September (before the start of the season, A1), after two months (the second week of November, A2), and two months after A2 (the third week of January, A3). All the assessments were performed under the same room and environmental conditions (place, time of day, order of tests application, temperature, and relative humidity, respectively, 22–24 °C and 55–65%) and by the same examiner. The players did not perform any other complementary training sessions during the period analyzed.

### 2.2. Participants

Seventeen elite women soccer players with a mean ± standard deviation age, height, body mass, and body mass index of 22.7 ± 6.3 years, 167.5 ± 5.6 cm, 60.7 ± 6.6 kg, and 21.6 ± 0.2 kg/m^2^, respectively, participated in this study. Their experience as professional soccer players was 4.7 ± 2.2 years.

We estimated the power of the sample size using a post hoc *F*-test: the within-group factor in a repeated-measures MANOVA, according to statistical method analyzed. The analysis featured 94.2% of actual power, with a total of 17 women soccer players with a *p* < 0.05 and effect-size for 0.6, using G-Power [16].

The players belonged to a team that participated in the Portuguese BPI League during the 2019/20 in-season. The inclusion criteria were regular participation in most of the training sessions (80% of the weekly training sessions) and the completion of at least half the matches in the first half of the season [17], while the exclusion criteria were injury, illness, sickness, and/or non-performance of all the assessments. Due to the exclusion criteria, only sixteen women soccer players participated in the present study. The field positions of the players in the study consisted of one goalkeeper, three central defenders (CD), three wide defenders (WD), three central midfielders (CM), four wide midfielders (WM) and three strikers (ST).

Despite the different characteristics of the soccer field players, the goalkeeper was included in the analysis, since all the data collected for this player were similar to the squad average and the players’ position values, and it was not detected as an outlier. All the participants were familiarized with the training protocols and the study design was carefully explained to the athletes. Written informed consent was obtained prior to the investigation.

A food frequency questionnaire to assess nutritional intake was applied over a 7 day period using a 24 h diet record, during the first week of the assessment 1 and during the last week of the assessment 3, in order for the players to verify their habits and food regimen routines.

The participants were instructed regarding portion sizes, supplements, food preparation aspects, and other aspects pertaining to an accurate recording of their energy intake. The records were reviewed for macronutrient composition and total energy intake [7]. All the participants were asked to maintain their normal diet throughout the study period.

The study was conducted according to the requirements of the Declaration of Helsinki and all the procedures were approved by the research Ethics Committee of the Polytechnic Institute of Santarém, Santarém, Portugal. All the subjects received their club’s medical approval to participate in the study and were instructed not to take any medication during the study.

### 2.3. Procedures

The data were collected in weeks with only one match, which means that the team typically trained three days a week (match day minus (-); MD-5; MD-4; MD-2). This approach was used in a previous study [17]. During the period analyzed, a total of 57 training sessions and 13 matches occurred. The 57 training sessions were divided into 19 speed endurance sessions (e.g., long sprints, repeated sprints), 19 aerobic high-intensity sessions (e.g., interval training, medium-to-large sized games), and 19 ball-possession games and team/opponent tactics sessions. Figure 1 presents the timeline of the study.

In order to produce more specific information regarding training and match load, rated perceived exertion (RPE) and the duration of training sessions and matches were collected and presented in Table 1 to quantify training load. The data are presented by squad average between the different assessments. On match days (MD) only the average data for starters were included.

### 2.4. Anthropometric and Body Composition Assessment

Based on previous recommendations, the anthropometric and body composition measures were obtained with the subjects dressed in light clothing without shoes [18,19]. The participants were further asked to remove all objects that could interfere with the bioelectrical impedance assessment. The participants’ weight and height were measured using a stadiometer with an incorporated scale (Seca 220, Hamburg, Germany) according to standardized procedures [20]. The body composition data were obtained with bioelectrical impedance analysis through Inbody S10 (model JMW140, Biospace Co, Ltd., Seoul, Korea), according to manufacturer’s guidelines [21,22] and the recommendations of a previous study [23]. Eight electrodes were placed on eight tactile points (thumbs, middle fingers and ankles of both hands and feet, respectively) to perform the multi-segmental frequency analysis. Next, a total of 30 impedance measurements were obtained at frequencies 1, 5, 50, 250, 500, and 1000 kHz, respectively, from different segments of the body, such as the right and left arms, trunk, and right and left legs, respectively. Moreover, three different frequencies (5, 50, and 250 kHz) were used to collect the 15 reactance, PhA measurements from the right and left arms, trunk, and right and left legs, respectively. The variables collected were: body fat mass (BFM), soft lean mass (SLM), fat-free mass (FFM), intracellular water (ICW), extracellular water (ECW), total body water (TBW), phase angle (PhA, 50 Khz), ECW/TBW ratio, and ECW/ICW.

The measurements were carried out in the morning [18,24], in a room with an ambient temperature and relative humidity of 22–23 °C and 50–60%, respectively, after a minimum of 8 h of fasting and after the bladder was emptied. The participants adopted a supine position with their arms and legs abducted at a 45° angle, and the right hand and foot dorsal surfaces were cleaned with alcohol. After a 10 min rest in a room without noise, eight electrodes were placed on the cleaned surfaces and the measurements were performed. The subjects did not exercise or ingest caffeine or alcohol during the 12 h prior to the assessment and they were only assessed if they were in the luteal phase of ovulatory menstrual cycles. Otherwise, they waited for more days, until they were in the luteal phase. All the assessments were performed by the same evaluator in order to minimize possible measurement errors [25].

### 2.5. Training and Match Load Quantification

Thirty minutes after the end of each training session and match, the players were asked to provide an RPE (0–10 scale) [26]. The players were prompted for their RPE individually using a custom-designed application on a portable computer tablet. They selected their RPE rating by touching the respective score on the tablet, which was then automatically saved under the player’s profile. This method helped to minimize factors that may have influenced the player’s RPE rating, such as peer pressure and replicating other players’ ratings [27]. Next, the s-RPE was calculated, as in our previous studies, through the multiplication of the session duration by the RPE [28,29].

### 2.6. Statistical Procedures

Descriptive statistics (mean ± standard deviation) were performed for all the measurements. All the variables were checked for normality and homoscedasticity, respectively, using the Shapiro–Wilk and Levene tests. The MANOVA with repeated measures was performed for the variables that obtained normal distribution to compare the three assessments, with s-RPE being used as covariate. The value of *p* ≤ 0.05 was established as significant and all the data were analyzed using SPSS version 22.0 (SPSS Inc., Chicago, IL, USA) for the Windows statistical software package. Furthermore, the change (%) was calculated between each comparison. The Cohen’s d effect-size (ES) was performed to determine the effect magnitude through the difference of two means divided by the standard deviation from the data, and the following criteria were used: <0.2 = trivial, 0.2 to 0.6 = small effect, 0.6 to 1.2 = moderate effect, 1.2 to 2.0 = large effect, and >2.0 = very large [30].

## 3. Results

Table 2 summarizes the participants’ characteristics by player position, while Table 3 showed comparisons between the three assessments for the squad average.

After performing ANCOVA with the session’s rated perceived exertion (s-RPE) as the covariate, no linear interaction was demonstrated between this variable and any of the other body composition variables (*p* > 0.05). Table 2 shows significant differences between A1 and A2 with moderate to very large effect, namely, BFM (*p* = 0.029; ES = 1.58), SLM (*p* = 0.018; ES = 1.24), FFM (*p* = 0.010; ES = 1.34), ICW (*p* = 0.007; ES = 1.41), TBW (*p* = 0.018; ES = 1.04), ECW/TBW (*p* = 0.002; ES = 10.00), and ECW/ICW (*p* = 0.022; ES = 3.33).

In addition, there was only a significant difference with very large effect between A2 and A3, for ECW/TBW (*p* = 0.001; ES = 3.33).

Finally, there were significant differences with large to very large effect between A1 and A3 for BFM (*p* = 0.029; ES = 1.87), FFM (*p* = 0.045; ES = 1.51), ICW (*p* = 0.049; ES = 1.68), ECW/TBW (*p* = 0.013; ES = 10.00), and PhA (*p* = 0.001; ES= 6.64).

## 4. Discussion

In this study, we aimed to identify changes in the body composition of elite women soccer players during in-season through BIA. Our main findings showed improvements in body composition, namely decreased BFM, increased FFM, and increased PhA; and a better fluid distribution was observed, especially from the first to the last assessment. However, no significant differences were noted between A2 and A3, except for ECW/TBW.

On one hand, BFM has been shown to exert a negative influence in athletes’ performance [5]. On the other hand, FFM has been associated with performance improvements [5]. In our study, the athletes showed a significant decrease in BFM and an increase in FFM. These results are similar to those reported in another study [31], which assessed athletes’ body composition in 5 time-points during the in-season. Regarding BFM, athletes presented mean values similar to those found by other authors [32] that assessed body composition changes pre-to-post-season in women soccer players. However, the authors found that the soccer players lost lean mass tissue over the competitive season that was not recovered during the off-season [32]. These results may be attributed to overtraining or negative energy balance.

Concerning PhA, the present female soccer players showed a mean value of 6.26 ± 0.11° in A1, 6.67 ± 0.31° in A2, and 6.99 ± 0.10° in A3. All these values are similar to those found in other studies conducted on women athletes or active young populations [33,34,35]. Furthermore, the values obtained in this study were slightly lower in A1 and A2 compared to those obtained in a study of healthy adult non-athletes [36]. Moreover, PhA has been related to cellular health and integrity [37]. For example, muscle injuries can cause a reduction in PhA which can provoke cell membrane disruption [38,39], which has also been related to body composition [33,40,41]. For instance, FFM is directly related with PhA [41]. Indeed, as FFM increased in these athletes, it seems plausible that PhA also increased. An improvement in PhA can be an indicator of good health and cellular integrity and functionality regarding the level of hydration [34]. Another application of PhA is related to cellular energy levels, so the low phase angle is consistent with an inability of cells to store energy, as well as being an indication of breakdown in the selective permeability of cellular membranes. A high PhA is consistent with large quantities of intact cell membranes and body cell mass [42].

Regarding TBW and its compartments, the importance of TBW and ICW in increasing performance in athletes is clear [7,8,9]. The increment in ICW and TBW in the present study is in line with a previous study that used resistance training in healthy and young adults [35]. In this regard, soccer is characterized by high intensity bouts of activities and movements. Glycogen is an essential substrate during high intensity sports [43]. Therefore, some explanations could be related to cellular hydration by increasing the glycogen storage, since glycogen features great osmotic power (each gram of glycogen is stored in human muscle with at least 3 g of water) [43]. These results are very important for athletes, since ICW content may stimulate pathways that increase protein synthesis [44,45]. ECW did not show any change during the in-season. Furthermore, the ECW/ICW ratio has been used as an indicator of fluid distribution in athletes [7,8,9,33]. Two recent studies [33,34] found values of 0.7 ± 0.1 in women athletes. In our study, the soccer athletes demonstrated mean values of 0.60 in A1, 0.59 in A2, and 0.59 in A3. Lower values of ECW/ICW have been found in athletes has and they have been associated with improved performance [7].

As mentioned earlier, when A2 andA3 were compared, no significant differences were found. These findings could be attributed to the increased training load in the beginning of the in-season that is generally found in soccer teams [46]. The higher training load resulted in body composition improvements in this early phase (between A1 and A2) that were followed by an adaptation in the second phase of the study (between A2 and A3), causing a maintenance of the body composition variables (considering that nutritional intake was controlled). This is important to highlight because in fact training load was higher between A2 and A3 without, however, changing any body composition variable.

A relevant finding that should be highlighted regards s-RPE. Through the analysis conducted in the present study, no interaction was observed between s-RPE and any body composition variables, which means that RPE can be dissociated from the physiological process through different psychological mechanisms [47]. As mentioned in previous studies, it seems that RPE was a simplification of the perceived psychophysiological exertion. Consequently, the use of this measure alone did not conclusively capture different sensations and experience of training sessions [47,48]. Furthermore, RPE was collected 30 min after the training sessions and that value included the entire session. This means that there could be some possible variation during training sessions in different exercises, as suggested by Ferraz et al. [48], that were not controlled in this study. This explanation may help to explain the non-interaction found regarding this variable in this study. It also reinforces the use of additional variables in training load monitoring, such as distances covered at different intensities, accelerations, decelerations, player load and metabolic power.

Scientific research on women soccer athletes is scarce [10,11,49], especially at the elite level, and to the best of our knowledge, this is one of the first studies to include several variables in order to assess anthropometric and body composition in elite women soccer players from a Portuguese BPI Ligue team. However, the sample size was derived from one team only and, therefore, future studies are required to generalize the findings.

Another interesting finding was related to the goalkeeper analysis, which showed that s-RPE and different body composition variables were similar regardless of player position and the squad average values. However, future studies are required to confirm this finding, since only one goalkeeper is an insufficient sample size from which to draw definite conclusions. Furthermore, more players for each position are required for an analysis across player positions.

Despite the importance of these results, and despite the use of tetra-polar and multi-frequency bioimpedance equipment, such as InBody S10, to assess body composition and fluid distribution, we should address, as a limitation of this study, the use of a non-considered reference method. Another limitation was the fact that it was not possible to make comparisons among athletes of different field positions, as this would reduce the sampling power. Finally, and despite the fact that no differences were found in nutritional intake, this assessment was performed through a questionnaire at two time points, which should be better addressed in future studies. Even so, this study represents the actual training routine followed by the specific team analyzed. Therefore, more research is needed with larger numbers regarding soccer players and teams over an all-season period.

## 5. Conclusions

Coaches, physicians, nutritionists, and exercise physiologists should ensure they provide gender-specifications for optimizing performance. This study highlights information on the essential characteristics of successful women’s’ soccer team performance at three time-points throughout the sport season. For instance, the study showed that although some players may have performed different field roles and positions, their body composition characteristics improved over the season, which reveals that nutritional habits were controlled and, consequently, the intensity of training and matches did not affect the body composition variables.

This study presents a report using body composition data and internal training load simultaneously, which can be used as a reference for better body composition, training load and performance management for coaches and their staff. However, we recommend that future studies include a full season and other training load measures, such as global positioning systems, to amplify the present findings.

## Figures and Tables

**Figure 1 ijerph-18-12023-f001:**
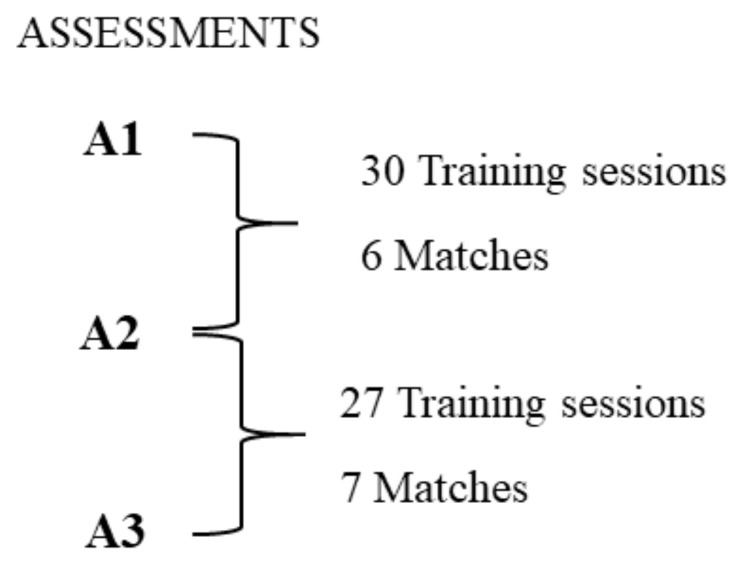
Timeline of the study. Legend: A1. Assessment 1; A2. Assessment 2; A3. Assessment 3.

**Table 1 ijerph-18-12023-t001:** Training and match RPE and duration between the three assessments.

Periods	Variables	MD-5	MD-4	MD-3	MD-2	MD-1	MD
	RPE (au)	5.5	5.4	X	4.8	X	6.2
Between A1 and A2	Duration (min)	87	85	X	77	X	72
	s-RPE (au)	478.5	459	X	396.6	X	446.4
	RPE (au)	6.1	5.5	X	4.1	X	6.5
Between A2 and A3	Duration (min)	85	85	X	90	X	90
	s-RPE (au)	518.5	467.5	X	369	X	585

A1. Assessment 1; A2. Assessment 2; A3. Assessment 3; MD. Match-day; MD-5. Match minus five days to the match. respectively for -4, -3, -2, and -1. RPE. Rated perceived exertion; s-RPE. Session rated perceived exertion; au. Arbitrary units; min. Minutes. X indicates day off.

**Table 2 ijerph-18-12023-t002:** Participant characteristics by player position in the three assessments.

Variables	Goalkeeper *n* = 1	Central Defender *n* = 3	Wide Defender *n* = 3	Central Midfielder *n* = 3	Wide Midfielder *n* = 4	Striker *n* = 3
**Assessment 1**						
Body weight (kg)	64.0	71.0 ± 2.0	54.3 ± 3.8	59.3 ±9.2	53.5 ± 8.7	57 ± 1.0
Body fat mass (kg)	15.3	18.7 ± 2.3	12.4 ± 1.4	14.1 ± 5.4	11.1 ± 4.5	8.1 ± 2.0
Soft lean mass (kg)	45.9	49.1 ± 2.1	39.3 ± 2.3	42.5 ± 3.6	39.9 ± 5.1	46.0 ± 2.9
Fat free mass (kg)	48.7	52.3 ± 2.2	41.9 ± 2.4	45.2 ± 3.9	42.5 ± 5.4	48.9 ± 3.0
Intracellular Water (L)	22.4	23.8 ± 0.9	19.1 ± 1.3	20.6 ± 1.9	19.3 ± 2.4	22.4 ± 1.3
Extracellular Water (L)	13.2	14.4 ± 0.8	11.4 ± 0.6	12.5 ± 1.0	11.7 ± 1.5	13.3 ± 1.0
Total Body Water (L)	35.6	38.2 ± 1.7	35.5 ± 1.8	33.1 ± 2.8	31.0 ± 4.0	35.7 ± 2.3
Phase Angle (θ. 50 Khz)	6.8	6.0 ± 0.3	6.3 ± 0.6	6.3 ± 0.5	6.0 ±0.3	6.4 ± 0.3
**Assessment 2**						
Body weight (kg)	67.0	69.3 ± 1.2	53.7 ± 3.2	58.0 ± 6.9	53.5 ± 7.9	57.0 ± 2.0
Body fat mass (kg)	15.8	14.1 ± 2.6	8.8 ± 4.2	10.2 ± 1.9	10.9 ± 3.7	6.7 ± 1.8
Soft lean mass (kg)	48.1	51.7 ± 2.1	42.0 ± 6.0	44.8 ± 4.9	39.9 ± 5.3	47.2 ± 3.5
Fat free mass (kg)	51.2	55.3 ± 2.3	44.9 ± 6.2	47.8 ± 5.1	42.6 ± 5.6	50.3 ± 3.7
Intracellular Water (L)	23.5	25.2 ± 1.1	20.5 ± 3.0	21.8 ± 2.4	19.4 ± 2.7	23.1 ± 1.7
Extracellular Water (L)	13.9	15.0 ± 0.6	12.1 ± 1.6	13.1 ± 1.3	11.6 ± 1.4	13.6 ± 1.1
Total Body Water (L)	37.4	40.1 ± 1.7	32.6 ± 4.6	34.8 ± 3.7	31.0 ± 4.1	36.7 ± 2.8
Phase Angle (θ. 50 Khz)	6.8	6.5 ± 0.6	7.8 ± 2.6	6.6 ± 0.7	6.2 ± 0.4	6.7 ± 0.2
**Assessment 3**						
Body weight (kg)	67	69.0 ± 2.6	53 ± 4.4	57.0 ± 6.2	53.8 ± 7.4	59.0 ± 1.7
Body fat mass (kg)	15.4	12.1 ± 3.6	8.0 ± 2.8	12.2 ± 3.6	9.4 ± 3.1	8.7 ± 1.7
Soft lean mass (kg)	48.4	53.3 ± 5.4	42.2 ± 2.6	42.0 ± 2.6	41.5 ± 4.1	47.2 ± 2.9
Fat free mass (kg)	51.6	56.9 ± 5.6	45.0 ± 2.7	44.8 ± 2.9	44.4 ± 4.5	50.3 ± 2.9
Intracellular Water (L)	23.7	26.1 ± 2.7	20.6 ± 1.4	20.4 ± 1.3	20.1 ± 2.0	23.1 ± 1.4
Extracellular Water (L)	13.9	15.4 ± 1.4	12.1 ± 0.6	12.2 ± 0.7	2.3 ± 1.3	13.5 ± 0.8
Total Body Water (L)	37.6	41.4 ± 4.1	32.7 ± 1.9	32.6 ± 2.0	32.3 ± 3.3	33.6 ± 2.2
Phase Angle (θ. 50 Khz)	7.4	7.1 ± 0.6	7.4 ± 0.5	6.9 ± 0.3	6.6 ± 0.2	6.9 ± 0.3

**Table 3 ijerph-18-12023-t003:** Comparisons between assessments by squad average (*n* = 17).

Variables	A1	A2	A3	Change % (A1–A2)	Change % (A2–A3)	Change % (A1–A3)
Body weight (kg)	58.74 ± 2.15	58.30 ± 1.97	58.30 ± 1.94	−0.8	0.0	−0.8
Body fat mass (kg)	13.11 ± 1.87 a	10.77 ± 0.94	10.38 ± 0.87	−21.7	−3.8	−26.3
Soft lean mass (kg)	42.87 ± 1.20 a	44.52 ± 1.44	44.91 ± 1.42	3.7	0.9	4.5
Fat free mass (kg)	45.63 ± 1.27 a.c	47.52 ± 1.53	47.92 ± 1.51	4.0	0.8	4.8
Intracellular Water (L)	20.79 ± 0.58 a.c	21.71 ± 0.72	21.88 ± 0.71	4.2	0.8	5.0
Extracellular Water (L)	12.53 ± 0.35	12.88 ± 0.41	13.00 ± 0.39	2.7	0.9	3.6
Total Body Water (L)	33.32 ± 0.93 a	34.59 ± 1.12	34.88 ± 1.09	3.7	0.8	4.5
ECW/TBW	0.38 ± 0.001 a.c	0.37 ± 0.001 b	0.37 ± 0.001	−2.7	0.0	−2.7
ECW/ICW	0.60 ± 0.003 a	0.59 ± 0.003	0.59 ± 0.004	−1.7	0.0	−1.7
Phase Angle (θ. 50 Khz)	6.26 ± 0.11 c	6.67 ± 0.31	6.99 ± 0.10	6.1	4.6	10.4

A1. Assessment 1; A2. Assessment 2; A3. Assessment 3; ECW. Extracellular water; ICW. Intracellular water; TBW. Total body water. The symbol a denotes significant difference between A1 and A2 (*p* < 0.05). The symbol b denotes significant difference between A2 and A3 (*p* < 0.05). The symbol c denotes significant difference between A1 and A3 (*p* < 0.05).

## Data Availability

The data presented in this study are available on request from the corresponding author.

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
