# Peer review of "In-Season Body Composition Effects in Professional Women Soccer Players"

_ijerph, 2021, doi:10.3390/ijerph182212023_

Round 1

Reviewer 1 Report

This paper addresses physical-bio changes among soccer athletes. The purpose was to analyse the anthropometric and body composition effects in professional soccer women players across the early and mid-competitive 2019/20 in-season. Seventeen players from a Portuguese BPI League team participated in this study. Although targeting an important topic, this paper needs major revisions aimed at addressing "areas of weakness" in the introduction, methods, results, and discussion. The arguments justifying the need for the study are not well developed, and thereby need to be advanced. Furthermore, the authors need to discuss power issues and comment on "why" soccer (convenience sampling?). It is important to ensure confidentiality, and thus remove any identifiable and personal information from the manuscript. Most importantly, I believe the discussion is very speculative at times. A native or highly fluent English writer should assist with grammatical issues.

ABSTRACT

Delete results data from the abstract.

INTRODUCTION

I think this is a worthwhile topic, which was relatively well covered. Nonetheless, it is essential to advance the argument/justification about the need for conducting this study. It is too short, you must to improve it.

METHODS

Do not disclose the participants' team and any sort of personal information. While it is important to make a case about their competitive level, it is essential to ensure confidentiality.

I would like to see a note about the apparent absence of a power analysis and a bit more elaboration on the convenience sampling strategy adopted (if in fact this type of sampling was used) (i.e., "why" was it necessary?).

Discussion

The discussion section lacks of a clear statement at the end before limitations are presented. What is the main result, what is the main highlight of the study?

I would like to see very specific directions for future research as related to the findings of the present study.

Reviewer 2 Report

Thank You for allowing me to review this paper.

In my opinion, this is an interesting study. Women's football is not as popular as men's. Much more studies focus on males than females. Therefore, this study adds some interesting new observations which state base for further investigations.

Abstract

25-26 In the abstract, the subjects' beginning body height, weight, and BMI should be provided.

36, the abbreviation "TL" should be explained as it appears the first time in the text.

Introduction

54-57 I agree with the statement from this paragraph; however, I think it should be slightly developed to indicate your study's importance and justification. I think study below-mentioned study should be helpful:

  Collins J, Maughan RJ, Gleeson M, et al. UEFA expert group statement on nutrition in elite football. Current evidence to inform practical recommendations and guide future research. Br J Sports Med. 2021;55(8):416. doi:10.1136/bjsports-2019-101961

https://bjsm.bmj.com/content/55/8/416.long

Material and methods

93-97 Could You provide some information about your subject experiences in professional football?

110-111 Dietary questionnaire should be described much broader. What kind of questionnaire was used? Was it any standardized tool? Or your authors'? – if yes, did You validate it?

How long last dietary control?

What data does it provide? The calory intake was controlled? Was it any differences in measured body morphology parameters associated with dietary intake?

 Was water consumption tracked?

Discussion

226-228 For example, muscles injuries provide reduction in PhA 226 as showed in previous studies showing cell membranes disruption [37, 38]. Also, it has 227 been related to body composition [32, 39, 40]. For example, FFM is directly related with 228 PhA [40]. For example (..). For example (…) It not sound good. I suggest you to change it.

Throughout the 3rd and 4th paragraphs, you should explain why there were no differences between measurement in  A2 and A3?

Conclusion

I think you should answer in this paragraph, how could coaches and players use your results? Or how your study could be enhanced and developed? What should be done next in this area?

Round 2

Reviewer 1 Report

no comment

Reviewer 2 Report

Manuscript is improved. All issues were properly adressed.